# Enteric Viral Co-Infections: Pathogenesis and Perspective

**DOI:** 10.3390/v12080904

**Published:** 2020-08-18

**Authors:** Heyde Makimaa, Harshad Ingle, Megan T. Baldridge

**Affiliations:** Division of Infectious Diseases, Department of Medicine, Edison Family Center for Genome Sciences & Systems Biology, Washington University School of Medicine, St. Louis, MO 63110, USA; heyde.makimaa@wustl.edu (H.M.); hingle@wustl.edu (H.I.)

**Keywords:** enteric virus, co-infection, viral pathogenesis, co-infection models

## Abstract

Enteric viral co-infections, infections involving more than one virus, have been reported for a diverse group of etiological agents, including rotavirus, norovirus, astrovirus, adenovirus, and enteroviruses. These pathogens are causative agents for acute gastroenteritis and diarrheal disease in immunocompetent and immunocompromised individuals of all ages globally. Despite virus–virus co-infection events in the intestine being increasingly detected, little is known about their impact on disease outcomes or human health. Here, we review what is currently known about the clinical prevalence of virus–virus co-infections and how co-infections may influence vaccine responses. While experimental investigations into enteric virus co-infections have been limited, we highlight in vivo and in vitro models with exciting potential to investigate viral co-infections. Many features of virus–virus co-infection mechanisms in the intestine remain unclear, and further research will be critical.

## 1. Introduction

Every year, millions of children are infected with viruses that target the gastrointestinal tract to cause acute gastroenteritis (AGE), which is the inflammation of the stomach or small or large intestine [1,2]. Indeed, approximately one billion episodes of diarrhea occur in children under five annually [3]. Symptoms accompanying AGE include malaise, abdominal pain and cramping, nausea, vomiting, and diarrhea, usually lasting 1 to 5 days, but occasionally up to 14 days [4]. RNA viruses, including rotavirus (RV), norovirus (NoV), sapovirus (SaV), astrovirus (AstV), and enteroviruses that spread via fecal–oral transmission, as well as DNA viruses, including enteric adenoviruses (AdV), are associated with AGE in immunocompetent and immunocompromised individuals of all age groups [5] (Table 1).

In the past, AGE was generally attributed to a single causative agent known to be associated with clinical symptoms. However, due to their high prevalence, exposure to multiple viruses at similar times can potentially occur in the same host, and these viruses may either infect simultaneously or within a short window of time. The outcomes of either infection may thus be influenced by the combined contribution of multiple agents. With increasing capacity to efficiently analyze samples for the presence of multiple putative pathogens using quantitative real-time PCR approaches, our appreciation that more than one virus may be present in an infected individual has grown. Enteric virus co-infections are commonly identified for all viruses linked to AGE [14] (Table 1). Despite our increased awareness of the prevalence of enteric viral co-infections, however, the mechanisms underlying viral co-infection have not yet been carefully explored. There are significant gaps in our knowledge regarding the dynamics between co-infecting agents and how these influence disease severity, immune responses, or vaccine efficacy.

Here, we will review what is known about the clinical aspects of enteric virus co-infection and discuss potential models to investigate viral co-infection in vitro and in vivo. We will first summarize what is known about enteric viral co-infection prevalence and its impact on health and disease, as well as how vaccine responses may be influenced by co-infection events. Second, we will present several of the possible cellular and molecular consequences of enteric viral co-infections. Finally, we will conclude by exploring past and current model systems used to study viral infections in the gut that could be applied to co-infections.

## 2. Virus–Virus Co-Infection Prevalence

By quantity and diversity, prokaryotic and eukaryotic viruses in the human intestinal microbiota likely outnumber their bacterial counterparts, with the size of the virome still remaining largely unknown [15,16,17]. In this review, we will predominantly focus on intestinal eukaryotic viruses as these are the known causative agents of AGE, but will briefly mention that co-infection relationships between bacteriophages play a major role in shaping complex microbial ecosystems [18,19]. Superinfection exclusion serves to limit subsequent phage infection of bacteria already having a prophage present, but in other contexts virus–virus interactions may serve to facilitate infections especially when the co-infecting phage are highly distinct (for example, ssDNA and dsDNA phage) [20]. While phage co-infection interactions have predominantly been explored in vitro or in environmental samples, new evidence suggests phage co-infection is also occurring in the human gastrointestinal tract, and thus the interrogation of these interactions is likely to be an important aspect of future microbiota research [21].

Recently, the metagenomic sequencing analysis of stool samples of several cohorts of healthy neonates revealed a stepwise viral colonization model wherein prophages may be the viral pioneers in the gut at one month after birth while, by four months, eukaryotic viruses became prominent [22,23,24]. The first several years of life are then a particularly dramatic period of co-infection, and the simultaneous detection of multiple enteropathogens is common, both associated with AGE and even in the absence of symptoms. As an example of prevalence in early life, a cohort of Dutch infants had at least one enteropathogen, including eukaryotic viruses and bacterial pathogens, detected in ~73% of samples tested, and of these, nearly half had two distinct pathogens detected [25]. Similarly, a longitudinal study conducted in two healthy British infants during their first year of life showed that a variety of human viruses, including AdV, AstV, RV, and multiple enteroviruses, were present during this period [26]. Indeed, 92% of the samples tested in this study were positive for one to five different viruses, and approximately half of the samples contained two or more viruses. Gut pathogens, including enteric viruses, are thus extremely common in early life, and virus–virus interactions, apart from being synergistically pathogenic, may play important roles in the development of the host–virome homeostatic relationship, as well as immune system education [15].

Risk factors for acquiring enteric co-infections include young age, daycare attendance, and households with more than three children [25], as well as contaminated drinking water and poor sanitation [27]. Children from high income regions often exhibit lower rates of co-infection than those from lower income regions. For example, using the same assay, one French study found a 7% co-infection rate, compared to a 77% co-infection rate observed in a cohort of Ghanaian children [28,29,30]. In one of the largest multi-site studies to look at pathogens associated with community diarrhea in developing countries, one enteropathogen was detected in 77% of diarrheal stools and 65% of non-diarrheal samples, and two or more pathogens in 41 and 29%, respectively, in the first two years of life [31]. Thus, an improved understanding of the consequences of these extremely common co-infection events is critical. Below, we will briefly describe the major enteric viral pathogens, and then will detail what has been discovered about viral co-infection frequency worldwide.

*Reoviridae*: Despite the successful implementation of vaccine programs against rotavirus (RV) starting in 2006, it continues to be the leading cause of viral diarrheal mortality in children under the age of 5 globally, causing approximately 200,000 deaths per year [32]. To date, there are two internationally licensed oral RV vaccines [33,34]: the monovalent Rotarix and the pentavalent RotaTeq. Additional newer-generation RV vaccines are licensed in individual countries [33]. RV group A is the most common cause of AGE in countries without effective RV vaccination programs, causing significant childhood morbidity and mortality. RV is generally understood to target the mature enterocytes of the small intestine, likely mediating diarrheal illness via both direct effects on the epithelium and indirect effects from host responses [35].

*Caliciviridae:* Norovirus (NoV) has emerged as a leading cause of pediatric gastroenteritis [36], in addition to its well-established role in epidemic diarrheal disease in adults in healthcare settings and cruise ships [37]. NoV GII strains are the most common viral cause of epidemic AGE worldwide, with GII.4 strains most frequently associated with person-to-person transmission, due to their extremely contagious nature and their rapid rate of genetic and antigenic evolution [38]. Indeed, NoV causes ~18% of all cases of diarrhea in outpatient and inpatient settings among all age groups [39]. NoV has been shown to infect both immune cells and enteroendocrine cells of the small intestinal epithelium in immunocompromised patients [40,41]. Though less widely appreciated as a cause of AGE, sapovirus (SaV) is another member of the *Caliciviridae* family that causes diarrheal illness similar to NoV, especially in young children and the elderly, and frequently closely follows NoV in prevalence [42,43]. Although there are several NoV vaccines in development and two in clinical trials (bivalent GI.1/GII.4 vaccine and monovalent GI.1 oral vaccine) [44,45], there are no licensed vaccines against NoV or SaV.

*Astroviridae:* Astroviruses (AstV) are causative agents of diarrheal disease particularly in children and immunocompromised patients. Few studies report AstV-mediated AGE in normal healthy adults, but most children are infected with AstV and develop antibodies to the virus early in life [46,47]. Classical AstV has eight serotypes (AstV1–8) that usually cause self-limiting infections but can spread systemically in immunocompromised individuals [48]. Some novel strains of AstV have also recently been associated with central nervous system infections, mostly in immunocompromised patients, but their association with AGE is not well-established [49,50,51,52]. The replication of human AstV in enteroids in vitro indicates a multi-cellular tropism, including both intestinal progenitor cells and mature enterocytes [53].

*Picornaviridae*: Enterovirus species A and B (enteroviruses and coxsackieviruses), and C (polioviruses (PV)) are prevalent enteric pathogens causing a range of diseases, including AGE. PV is an enteric virus transmitted through the fecal–oral route, which during the pre-vaccine era could spread from the gut to the central nervous system to cause poliomyelitis, a disease leading to temporary or permanent paralysis [54,55]. Following the launch of the Global Polio Eradication Initiative in 1988, global vaccination programs led to the eradication of the indigenous wild PV type 2 in 2015 and type 3 in 2019 [56]. PV vaccines include an inactivated PV (developed by Jonas Salk) administered via injection [57] and a live attenuated PV (developed by Albert Sabin) administered orally—the oral polio vaccine (OPV) [58]. Both vaccines show high efficacy, but OPV is considerably easier to deploy in low-resource settings around the world and confers longer-lasting community immunity [59]. However, in rare cases, OPV can revert to its pathogenic variant and provoke poliomyelitis [60], and recent vaccine-derived PV outbreaks have been documented in several countries, highlighting the importance of continued PV research [61]. Other human enteroviruses, including enterovirus-A71, coxsackievirus-A6 and coxsackievirus-A16, cause hand, foot, and mouth disease (HFMD), predominantly in children aged 1 to 3 years old worldwide [62]. Enterovirus-A71 is closely associated and co-circulates with coxsackievirus-A16, causing seasonal outbreaks of HFMD in Central China, with reported co-infection in ~2% of cases [63]. In general, co-infection between enterovirus-A71 and other enteroviruses during HFMD outbreaks are associated with more severe symptoms [63,64,65]. While human enteroviruses have not classically been associated with AGE, there is increasing appreciation that many of these viruses can be associated with pediatric diarrhea [65,66,67,68].

*Adenoviridae*: Adenoviruses (AdV) are non-enveloped DNA viruses known to cause conjunctivitis, upper and lower respiratory disease, and AGE. AdV can be transmitted via respiratory droplets, fomites, and fecal–oral transmission, and predominantly infects the respiratory and gastrointestinal epithelia [69]. Epidemiological data indicate that the majority of AdV infections occur in children less than 5 years of age, but AdV epidemics are also common in adults. AdV genotypes 40 and 41 are commonly associated with AGE in pediatric populations worldwide [70,71,72,73,74]. The intestinal tract also appears to be a common site of AdV reactivation under conditions of immunosuppression [75]. In patients with congenital immunodeficiencies, AdV can cause disseminated and other lethal disease [76,77,78], whereas in acquired immunodeficiency associated with HIV infection, AdV infections are more commonly associated with AGE [79]. Although AdV infection in the respiratory tract is well characterized, gastrointestinal infection is less well-understood in terms of cell tropism, entry mechanism and intestinal immune responses [75]. AdV is prevalent in intestinal biopsies from healthy individuals, with co-infection by multiple strains commonly detected. AdV DNA has been found in lamina propria lymphocytes, but the full cellular tropism of human AdV remains unclear [80].

*Coronaviridae*: Coronaviruses have been previously considered as possible rare causes of AGE in infants [11]. However, the end of 2019 witnessed the beginning of an ongoing global pandemic from novel severe acute respiratory syndrome coronavirus 2 (SARS-CoV-2), with nearly 19 million cases and >706,000 deaths to date (as of 6 August 2020), dramatically increasing the awareness of these viral pathogens. SARS-CoV-2 uses receptor protein ACE2 for entry, which is expressed on lung epithelium but also on enterocytes in the gastrointestinal tract [12,13]. Infection is often associated with gastrointestinal symptoms including diarrhea, and SARS-CoV-2 viral RNA can be detected in fecal samples and viral antigens identified in intestinal tissues of COVID-19 patients [81,82,83]. In addition, there is mounting evidence for the SARS-CoV-2 antigen in the intestines of animal models, such as macaques and ferrets [84,85]. Although SARS-CoV-2 infection exhibits milder clinical symptoms in children as compared to adults, prolonged fecal shedding has been observed in pediatric cases [83,86]. It has already been suggested that SARS-CoV-2 infection may prevent co-infection by other respiratory viruses [87]. While it is still unclear if SARS-CoV-2 exhibits fecal–oral transmission [88,89], the epidemic has raised important questions about this novel pathogen in the intestine and whether it may be an important future consideration in enteric viral co-infections.

### 2.1. Co-Infection in Acute Gastroenteritis (AGE)

Though co-infections are more commonly thought of as viruses from different viral families mediating simultaneous infection, they can, of course, also include concurrent infections by multiple viral strains or species of the same viral genus. An investigation by the Rota-net Italy Study Group demonstrated that mixed infections with two or more RV strains were present in 7.6% of their samples, with G1 + G9P(8) strains representing the most common mixed infections in children with AGE [90]. In a study investigating the extensive RV genomic variation during chronic infection in immunocompromised children, patients unable to clear their initial RV infection were subsequently co-infected with other circulating RV strains leading to prolonged life-threatening diarrhea [91]. Mixed co-infections with different NoV genogroups have also been reported, as young children may experience multiple and often repeated NoV infections with remarkable genetic diversity [92,93,94]. Mixed enterovirus co-infections have not been frequently detected in AGE [68,95], though they have in HFMD [96], but as enteroviruses are now increasingly appreciated as causative agents for AGE, increased testing may also increase co-infection detection rates. Continuously improving techniques for carefully genotyping enteric viruses in clinical samples will likely continue to reveal frequent co-infections by closely-related viruses.

During epidemic AGE outbreaks, such as those that occur from contaminated water, co-infection with multiple unrelated viruses can be common. For example, in a pediatric AGE outbreak in Finland, in which drinking water was contaminated by sewage, RV was detected in 66% of cases, calicivirus including either NoV or SaV in 62%, and both in 40% [97]. Similarly, 23.2% of samples were positive for multiple enteric viruses in two outbreaks of AGE in Mumbai [98]. Even in community AGE, however, co-infections make up a moderate proportion of cases and are likely under-reported as some etiologic agents may not be tested for across all studies.

In analyses pre-dating the RV vaccine, one study conducted in Spanish children found that ~5% of the cohort showed mixed infections associated with acute diarrhea. Virus–virus co-infections were more frequent, with RV–AstV and RV–AdV being most common, than virus–bacteria co-infections [99]. Similarly, a study interrogating the prevalence of viral infections among children with AGE in France found co-infections in 50/299 positive samples (16.7%), wherein 94% of the cases were combinations of RV with NoV, AstV, or AdV in children under the age of 15 months [100].

The introduction of the RV vaccine altered the prevalence of RV infection, but it continues to be frequently detected during AGE. A study of enteric RNA virus co-infections in diarrheic children and adults in Southwestern Canada found that 2.8% of total RNA virus-positive samples showed combinations of NoV GII-SaV, NoV GII-RV, NoV GII-AstV, SaV-AstV, and SaV-RV at varying percentages [101]. A similar study focused on pediatric AGE in Spain identified more than half of children as being infected with at least one virus, including AdV, RV, SaV, AstV, and NoV, with co-infections detected in 21% of cases [102]. In Turkish children under the age of 5, RV was present in about one-third of AGE samples, and co-infections were found in 10.4% of samples with RV-NoV and NoV-AdV as the most frequently observed [8]. In China, RV and NoV-GII co-infections were reported in 14% of children suffering from severe diarrhea [39]. A study conducted to determine the agents associated with AGE in children in Nepal reported that nearly all children tested had at least one enteric viral agent detected, and approximately half were co-infected with RV and enterovirus [27]. Overall, these studies support that in healthy children and adults experiencing AGE episodes, between ~3–50% may have more than one viral pathogen present. Some reports indicate that RV co-infections, at least in healthy children, are not associated with greater symptom severity than single RV infection [25,27], while another reported that co-infections with NoV may lead to more severe AGE with enhanced comorbidities, such as vomiting and fever, compared to NoV alone [94]. Further clinical studies exploring the effects of co-infection on symptom severity are needed to better clarify the relative contribution of co-infection to disease outcomes.

Even less clear, however, are the impacts of co-infection in immunocompromised patients. Immunocompromised patients have been reported to have only slightly higher prevalence of individual enteric viral pathogens as the general population [103,104], but may experience both more symptomatic and prolonged infections [103,105]. A potential consequence of prolonged infection in immunocompromised patients and the elderly may be the intra-host evolution of genetically distinct viral populations and viral spread in nursing homes [106,107,108]. A study comparing HIV-positive and HIV-negative children reported a higher prevalence of AstV and enterovirus co-infection in HIV-infected children, with multiple strains of AstV detected in HIV-infected children with or without AGE [109]. An investigation in HIV-seropositive and -seronegative children with diarrhea in Brazil reported a significantly higher rate of co-infections with >1 enteric virus including NoV, AdV and human bocavirus in HIV-1 seropositive children [110]. There have also been a number of case reports of severe co-infections in immunocompromised individuals, such as those of AdV and coxsackievirus detected in a child with a primary immunodeficiency with persistent severe diarrhea [111], or AdV and NoV co-infection reported in a patient diagnosed with chronic myeloid leukemia undergoing stem cell transplantation [112]. Certainly, immunocompromised patients with AGE are at increased risk of complications and prolonged hospitalizations. The additional study of co-infections in both immunocompromised and elderly cohorts may help to provide insights into mechanisms related to the immune regulation of enteric viruses.

### 2.2. Co-Infection in Asymptomatic Individuals

Both human gut virome studies and investigations specifically testing for viral pathogens have shown that individuals with detectable enteric virus loads in stool may be free of clinical symptoms or disease pathology [15,16,26,113]. The long-term shedding of AstV, RV, AdV, and enteroviruses was detected in healthy infants in the UK from 1 up to 12 months without any clinical symptoms for AGE [26]. Asymptomatic infection and prolonged shedding of NoV is also common [36,92,114].

While there are fewer studies that have examined viral co-infection in asymptomatic individuals, those that have support frequent enteric viral co-infection in the absence of AGE. Up to 17 different enteric viruses were detected in fecal samples from children without diarrhea in Bangladesh, and among the positive samples, 53% demonstrated co-infection with multiple viruses [115]. Similarly, in a study in Cameroon, 53.7% of fecal specimens sampled from asymptomatic healthy children contained at least one identifiable enteric virus, with co-infections detected in 35.2%, almost all including NoV [116].

While co-infections are thus prevalent even in the absence of symptoms, the literature does suggest that, in matched cohorts, co-infection rates tend to be higher in cases of AGE than in asymptomatic controls. The analysis of samples from a Japanese daycare revealed that 1.6% of stools from asymptomatic children exhibited viral co-infections, whereas viral co-infections accounted for 5.7% of AGE cases [117]. Similarly, in southwest China, RV-NoV GII co-infection was found in 1.1% of asymptomatic children tested, as compared to 4.4% of samples from AGE cases [39]. Continued sampling of asymptomatic pediatric cohorts in diverse geographic locations will be important to help more fully define the worldwide prevalence of viral co-infections in the absence of symptoms.

### 2.3. Co-Infection Effects on Vaccines and Treatments

As discussed above, vaccines are currently available for PV and RV, but not for other enteric viruses associated with AGE. These vaccines are administered in early childhood, which is also the most common period for viral co-infections to occur. The inhibitory effect of human enteroviruses on OPV efficacy has been recognized for many decades, and indeed has contributed to attempts to vaccinate children in months when these viruses were circulating at lower levels [118,119]. A systematic review supports that concurrent enterovirus infection is consistently associated with decreased seroconversion for some types of PV, and in general that concurrent diarrhea which could be attributed to different enteropathogens is associated with decreased per-dose seroconversion overall [120]. Recently acquired enterovirus infections seem to have more dramatic effects in preventing immune response development than persistent infections [121]. In addition to enterovirus infections inhibiting effective immune responses to OPV, there have been more recent studies suggesting these may also impair effective RV vaccine responses. Diminished RV IgA and failure to seroconvert after RV vaccination were associated with concurrent enterovirus infections in a cohort of infants in urban Bangladesh [122]. Importantly, the co-administration of OPV and RV vaccines themselves have also been suggested to inhibit effective vaccine responses [123,124].

While some recent studies have suggested that co-infections with bacterial or viral enteropathogens do not alter RV vaccine efficacy in a statistically significant manner, across these studies there is a trend for an 8–11% increase in vaccine efficacy against severe RV diarrhea in children who were not co-infected [30,125,126]. Thus, it remains possible that viral co-infections are an important contributor to decreased vaccine efficacy in settings with high enteropathogen burdens. Further work will be needed to fully clarify the impact of enteric viral infections on the induction of effective protective responses with vaccination, and will continue to be an important consideration as vaccines are developed for other enteric viruses, such as NoV.

While most enteric viruses causing AGE do not have specific treatments available, an important note is that frequently diarrheal episodes are treated with antibiotics which have no efficacy against viral pathogens. In the MAL-ED multi-site study, almost half of all diarrheal episodes were treated with antibiotics. Analysis later showed that 91.7% of these antibiotic courses were inappropriate [3]. Thus, improved recognition and detection of the enteric viruses that mediate AGE, as well as an appreciation that multiple viruses may be present to drive diarrheal disease, is increasingly important to limit unnecessary administration of antibiotics.

## 3. Cellular and Molecular Consequences of Co-Infection

Despite frequent reports of viral co-infections, their consequences and underlying mechanisms are poorly understood. Here, we discuss potential consequences of co-infections, suggestions which will need to be bolstered by careful future studies (Figure 1).

### 3.1. Viral Recombination or Reassortment

Co-infections with related strains facilitate viral recombination, which can occur via homologous or non-homologous recombination, or reassortment. Improvements in viral genomic sequencing methods have facilitated the detection of recombinants, leading to accelerated detection of these events over the past decade. Enteroviruses have been frequently observed to recombine to permit the emergence of new viral strains [127,128], and indeed OPV and circulating enteroviruses also have the capacity to generate recombinants with enhanced pathogenicity and fitness [129]. Recombination events are also frequent in human and animal AstV [130] and NoV [131]. Because RV is a segmented RNA virus, a concerning outcome for co-infection is reassortment of the viral genome that can facilitate the emergence of novel epidemic strains [132]. While there has been less evidence for AdV recombination during intestinal infection, it is certainly a possible outcome considering the frequently persistent presence of these viruses in the gastrointestinal tract [75]. Cross-family viral recombination events are much less likely, but certainly not impossible [133].

### 3.2. Virus Aggregates

Recent studies have revealed that multiple enteric eukaryotic viruses, including RV, NoV, and PV may be transmitted in aggregates or viral clusters [134,135]. Groups of assorted virions, often containing mutagenized genomes, are replicated and transmitted cell-to-cell in vesicle-like structures [135,136,137]. For aggregation to occur, virions must be able to accumulate and be transmitted from infected cells to neighboring cells with minimal interference. Aggregation may offer viruses multiple advantages, including an increased cellular multiplicity of infection [138], and greater opportunity for complementation or recombination to restore fitness [135]. Gut bacteria have been shown to facilitate the aggregation of PV, and this could be a mechanism of the bacterially-mediated enhancement of infection by other enteric viruses [139]. Fundamentally, aggregates serve as an important and newly-appreciated mechanism for the co-infection of multiple related virions. Whether aggregates have the potential to include multiple unrelated viruses, and their role in the cellular consequences of co-infections, remains to be seen.

### 3.3. Viral Interference

Viral interference is a phenomenon first recognized in the 1950’s, wherein a cell infected by a virus is resistant to a second super-infecting virus, mediated by interferons (IFNs) [140]. This protection is also conferred to neighboring cells. Viral interference has also been proposed as an important mechanism by which enterovirus co-infections limit OPV and RV vaccine efficacy [120]. While few studies have interrogated enteric viral co-infections in mice to date, we recently uncovered viral interference as an unexpected outcome in immunocompromised mice. We observed that chronic murine AstV infection in severely immunodeficient mice protected mice from both persistent murine NoV and murine RV infection [141]. This protection was transferable to other immunodeficient mice and was associated with the upregulation of IFN-lambda (IFN-λ) in the gut, which in turn protected the mice from infection by other enteric viruses [141]. Further details of this intrinsic defense mechanism remain to be uncovered, as do the nuances of differential viral strain effects, cell type-specific IFN-λ expression, and positive and negative regulation of heterologous antiviral responses. It is likely this is an important mechanism regulating viral outcomes in the intestine.

## 4. Model Systems to Study Virus Co-Infections

While mechanistic explorations of viral co-infections in the intestine have been limited, there are several useful model systems available to explore the pathogenesis of individual viruses which could be applied to co-infection studies (Figure 2). Here, we describe these models, and suggest that these may permit us to address remaining fundamental questions: How does co-infection alter the lifecycle, replication, and tropism of individual viruses? Do individual viruses promote or prevent infection by subsequent viral agents? Do co-infections ultimately alter disease outcomes? The use of animal models and in vitro tools will be key to dissecting the molecular underpinnings in enteric virus co-infections.

### 4.1. Non-Human Primates

Non-human primates (NHPs), including Vervet monkeys (*Chlorocebus pygerythrus*), Rhesus macaques (*Macaca mulatta*), and pigtailed macaques (*Macaca nemestrina*) have been used extensively to study enteric viral infections [142]. NHPs share many anatomical, immunological, and physiological similarities with humans and thus are excellent models to recapitulate the pathogenesis of enteric virus infections and/or co-infection in humans [143]. NHPs are susceptible to some human enteric viruses, making studies of human viral co-infections potentially tractable [144,145]. Indeed, many simian viruses isolated from NHPs in the wild or in captivity, including simian RV and rhesus enteric caliciviruses, are also important pathogens for in vitro and in vivo studies, such as viral receptor identification and determination of adaptive immune responses [146,147,148]. In fact, research in simian RV led to the formulation of the first commercial vaccines against human RV [148,149]. Virome studies in NHPs suggest that, especially when immunocompromised by infection with simian immunodeficiency virus, natural enteric viral co-infections may be common [150,151,152]. However, experimental co-infection with enteric viruses remains a relatively unexplored area.

### 4.2. Pig Models

Gnotobiotic pigs serve as a useful model to study human enteric viruses, as similarities to human gastrointestinal physiology and mucosal immune development provide significant advantages over other model systems [153,154]. The capacity to introduce defined microbial consortia and/or study the effects of infection in the absence of the microbiota also add a layer of utility to these models. Piglets have been used extensively to study human RV and NoV infection, as they exhibit diarrheal disease, gastroenteritis, and fecal viral shedding, and have been useful for antibody response studies [153,155]. For example, the detection of human NoV capsid protein in small intestinal enterocytes of infected pigs associated with increased apoptosis and epithelial barrier disruption may offer insights into diarrheal disease observed in humans [156]. In the last few decades, multiple porcine enteric viruses that cause acute diarrhea, vomiting, dehydration, and high mortality in neonatal piglets have been identified, including *Coronaviridae* family members, porcine epidemic diarrhea virus and porcine deltacoronavirus, as well as porcine NoV similar to human NoV GII strains [157,158,159]. These viruses show fecal–oral transmission similar to human enteric viruses, providing an opportunity to investigate the nuances of viral transmission and define factors controlling clinical outcomes [160]. Recently, enteric virus co-infections were detected in the swine industry in China wherein piglets with diarrhea showed higher rates of coinfections with porcine epidemic diarrhea virus and porcine kobuvirus [161]. Thus, pigs have substantial potential as models to study enteric virus co-infections.

### 4.3. Mouse Models

In a compelling survey of the gut virome to investigate whether house mice living in close proximity with humans in residential buildings carry human pathogens, multiple viruses from the *Parvoviridae*, *Picobirnaviridae*, *Astroviridae*, *Genomoviridae*, and *Circoviridae* families were identified [162]. Although this study did not find any human viral pathogens, it highlighted a diverse viral ecology consisting of various enteric viruses in the mouse gut, analogous to the enteric viral diversity observed in humans.

Laboratory mouse models (specific knockouts, immunodeficient, transgenic, etc.), which can be maintained free of some of these enteric pathogens, have been highly instrumental in studying enteric viral infections including pathogenesis and immune responses. Mice are easy to manipulate and can recapitulate many of the human enteric viral–host relationships [163]. Human NoV GII.4 has been shown to replicate in a BALB/c Rag-γc-deficient mice, suggesting that some human enteric viruses may directly infect small animal models to facilitate study [164], though the further development of mice susceptible to human enteric virus infection is an important target. The discovery of murine NoV in immunodeficient mice provided the opportunity to study NoV in a natural host and contributed immensely to the understanding of the cellular and molecular mechanisms of NoV pathogenesis [165,166,167,168,169]. While murine NoV does not cause diarrheal illness in adult mice, it was recently reported to induce diarrhea in some neonatal models [170]. Comparably, murine RV causes both diarrhea in neonatal mouse pups and spreads among littermates [171,172]. Recently, murine AstV were discovered in immunocompetent, as well as in chronically infected immunodeficient, mice [163,173,174]. These murine AstV generally cause asymptomatic infection and minimal pathology [163]. While natural transmission of these viruses is fecal–oral, administration of virus via alternate routes permits comparative studies and can provide useful insights into pathogenesis and antiviral mechanisms [166,175]. Importantly, studies using murine viruses have revealed that viral strains differing by only a few amino acids may have dramatically different in vivo outcomes, in cellular tropism, duration of infection, and in interactions with the host innate and adaptive immune systems [176,177,178]. These observations help to inform our understanding of strain-specific differences in human viruses as well. Thus, although mouse viral models may not mirror all clinical features of human enteric infections, they are useful in identifying key determinants of viral regulation in vivo.

Overall, there is a clear need, and opportunity for, the exploration of enteric viral co-infections in mouse models, in order to interrogate infection outcomes for individual viruses as well as innate and adaptive immune responses and how these are governed by the presence of multiple viruses during infection.

### 4.4. Drosophila melanogaster

Insect models, including the fruit fly *D. melanogaster*, can be used to study oral infection by bacterial and viral pathogens in well-characterized, fast-growing, and readily genetically-manipulatable hosts [179,180,181]. Drosophila C virus, related to mammalian enteric picornaviruses, and Flock House Virus, a persistent RNA virus, are natural enteric viral infections in *D. melanogaster*. Microbiota regulation of antiviral immunity can restrict enteric viral infections in *D. melanogaster* [182], supporting that multiple aspects of enteric viral infection in higher organisms can be effectively modeled in flies. Further, *Drosophila* can be infected orally by human arboviruses, permitting the exploration of both fly and human virus co-infections [183]. Taken together, *Drosophila* has important promise as a model for enteric viral co-infections.

### 4.5. Caenorhabditis elegans

The nematode *C. elegans* is the model organism with the most biologically simple intestine, comprised of just 20 clonal enterocytes [181]. *C. elegans* has a number of advantages, as it has a rapid generation time and can be efficiently screened using forward- and reverse-genetic, as well as drug or small molecule screening approaches. It has been instrumental in the discovery of multiple molecular pathways, such as RNA interference [184]. There are several natural enteric viruses that have been recently isolated from wild-caught nematodes, including Orsay virus, facilitating the study of host–virus interactions [185]. Further, *C. elegans* are potentially amenable to infection with non-natural nematode viruses via microinjection [186]. *C. elegans* is thus a potential model to identify host-pathogen interactions during enteric virus co-infections.

### 4.6. Immortalized In Vitro Cell Lines

Caco-2 and HT-29 lines, derived from human colorectal adenocarcinomas, are the most widely used in vitro immortalized cell systems to study mechanisms underlying host–pathogen interactions in the intestine [187,188]. The Caco-2 cell line becomes highly polarized when subjected to optimal conditions and can thus differentiate to resemble small intestine-like cells or remain undifferentiated as large intestinal cells [188,189]. This cell line has been used to cultivate a number of enteric viruses, including human NoV [190,191], human RV [192,193] and human AstV [194,195], with some success. Interestingly, Caco-2 cells have been used to perform limited co-infection studies, revealing that mixed virus infection with AstV or enterovirus interferes with RV replication [196]. Much like Caco-2 cells, HT-29 cells have been employed extensively for studies of different aspects of intestinal epithelial cell biology [197,198]. HT-29 have been used for human AdV-40 and -41 [199] and a variety of RV studies [200,201]. These cell lines thus offer highly tractable systems to interrogate virus–virus co-infections in intestinal epithelial cells.

### 4.7. Intestinal Organoids and Enteroids

In the last decade, pioneering work from several groups has demonstrated the capacity of pluripotent intestinal stem cells to develop into complex intestinal structures in cell culture, closely mimicking their in vivo counterparts. Two types of intestinal organoids are used for studying enteric viruses—organoids derived from pluripotent stem cells, known as human intestinal organoids (HIOs), or organoids derived from intestinal crypts grown in three dimensional cultures, known as human intestinal enteroids (HIEs) [202,203,204,205].

Both HIOs and HIEs are widely used to study enteric viral pathogenesis and innate immune responses in the intestinal epithelium. HIOs support the replication of clinical isolates of RV, and HIEs have been used to study host restriction and innate antiviral responses to RV [206,207]. Multiple human NoV strains have been successfully cultivated in HIEs derived from human intestinal biopsies [208], as have human AstV [53]. HIE models have also been pivotal in investigating antiviral signaling responses to enteroviruses and coxsackieviruses [209,210]. HIOs and HIEs thus offer efficient systems to cultivate previously non-cultivable human enteric viruses, and thus may be powerful systems to investigate enteric virus co-infections.

## 5. Conclusions and Future Directions

Single pathogens have been classically understood to be the etiologies of AGE. As the evidence for virus–virus co-infections grows, however, it is evident that the causative agents for this disease are often co-infecting viruses, and indeed that these co-infections may be occurring even in the absence of symptoms. However, our understanding of the consequences of enteric virus co-infection is still in its infancy. There is a great remaining need for careful exploration of the effects of co-infection on symptom severity and duration, as well as development of protective immunity, in clinical cohorts.

Notably, the frequency of co-infection, and the identity of implicated viruses, can be drastically altered with the introduction of immunization programs. For example, PV and RV vaccination programs dramatically reduced infection rates in communities worldwide. Thus, as vaccines are developed for other enteric viral pathogens, there may continue to be robust shifts in viral co-infection prevalence. We are also now developing an increasing appreciation that the commensal microbiota, diet, and host genetics may have complex regulatory effects on enteric viruses, all of which may variably contribute to disease outcomes from infection as well. Thus, there remains much to be discovered in terms of physiological effects of co-infections and other host factors on disease severity, immune defense mechanisms and vaccine responses. Although the different models discussed in this review have been used to study the pathogenesis of individual enteric viruses, our knowledge regarding enteric viral co-infections lags far behind. We suggest that these models could be used as a toolkit for targeted studies exploring clinically relevant enteric virus co-infections, thereby allowing us to carefully dissect the individual contributions of the many complex factors that control these pathogens.

## Figures and Tables

**Figure 1 viruses-12-00904-f001:**
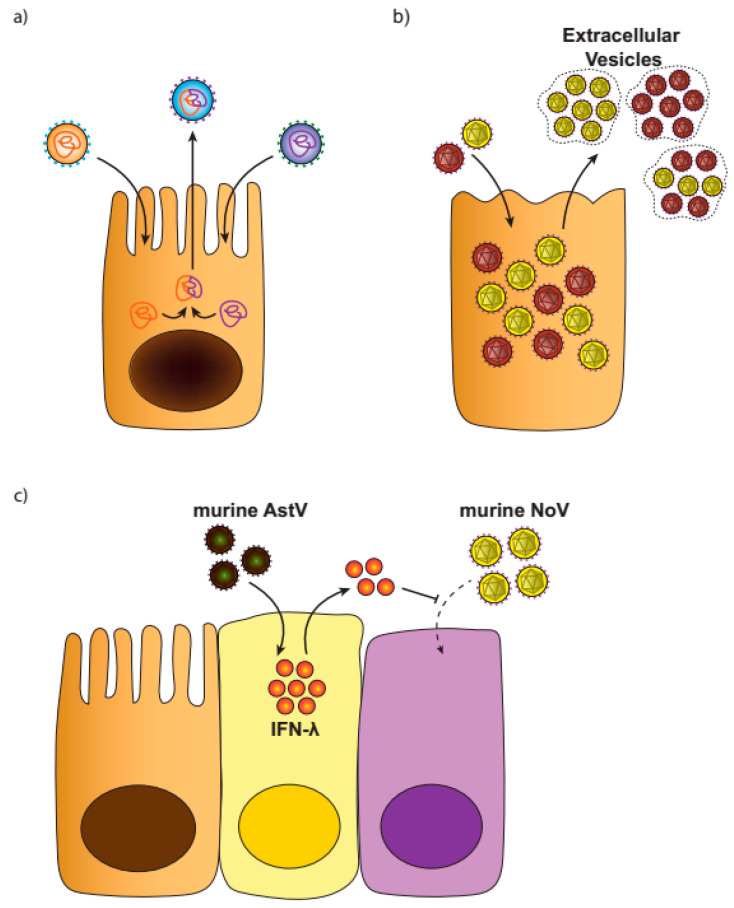
Potential consequences of enteric viral co-infections. (**a**) Co-infection of multiple viral strains in a single cell facilitates the recombination and generation of novel strains. (**b**) Extracellular vesicles containing multiple virions of a single virus type, one means of co-infecting new cells. Whether co-infection with unrelated viruses could lead to co-packaging is unknown. (**c**) An example of enteric viral interference, wherein murine AstV infection inhibits murine NoV infection of tuft cells via IFN-λ production.

**Figure 2 viruses-12-00904-f002:**
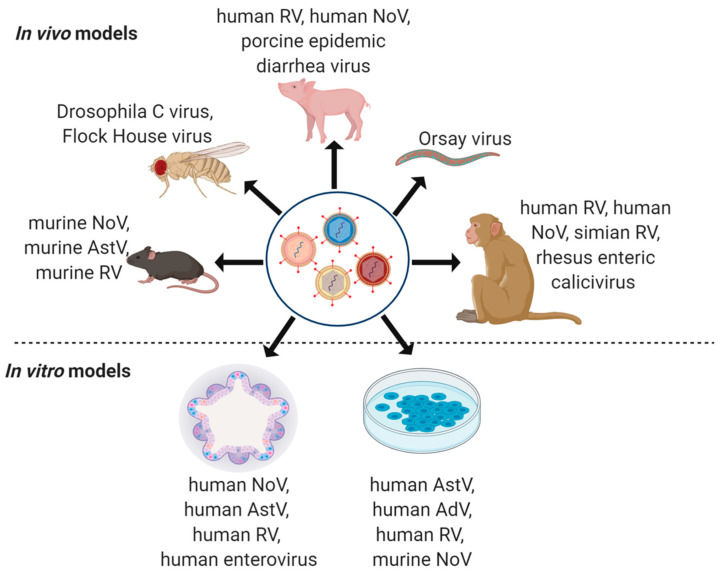
In vivo and in vitro co-infection study models. In vivo models for enteric virus studies include mouse (murine NoV, AstV, and RV), fly (Drosophila C virus, Flock House virus), pig (human RV, human NoV, and porcine epidemic diarrhea virus), *C. elegans* (Orsay virus) and non-human primate (human RV and NoV, simian RV, and rhesus enteric caliciviruses) models with physiological experimental read-outs. In vitro models that could be used for enteric co-infection studies include organoids/enteroids and immortalized cell lines for detailed analyses in simplified systems. Illustration created using BioRender (BioRender.com).

**Table 1 viruses-12-00904-t001:** Summary of major enteric viral pathogens, many of which have been frequently implicated in virus–virus co-infection events.

Classification	Genome	Major Agents	Disease	Prevalence	Reference
Class I *Adenoviridae*	dsDNA 30–40 kb	Adenovirus (AdV) serotypes 40, 41 (species F)	Gastroenteritis	15% in children and 1.5 to 5.4% in adults	[5]
Class III *Reoviridae*	dsRNA 16–27 kb	Rotavirus (RV)	Gastroenteritis	In children <5: pre-vaccine 40% post-vaccine 19%	[5,6]
Class IV *Astroviridae*	ssRNA (+) 6.8–7.9 kb	Astrovirus (AstV)	Nosocomial or epidemic diarrhea	~5% in children with AGE, 3–9% in hospitalized patients with diarrhea	[5,7]
Class IV *Caliciviridae*	ssRNA (+) 7.4–8.3 kb	Norovirus (NoV), Sapovirus (SaV)	Epidemic diarrhea	18% in all AGE cases, as well as 24% in community	[5,8,9]
Class IV *Picornaviridae*	ssRNA (+) 7.1–8.9 kb	Enterovirus species A and B (Enteroviruses and Coxsackieviruses), species C (Polioviruses)	Gastroenteritis, Hand, foot, and mouth disease (HFMD), Poliomyelitis	~5% in children with AGE; ~4% in children with HFMD	[10]
Class IV *Coronaviridae*	ssRNA (+) 27–30 kb	Severe acute respiratory syndrome (SARS-CoV) and severe acute respiratory syndrome coronavirus 2 (SARS-CoV-2)	Acute diarrhea, severe acute respiratory syndrome, and coronavirus disease 2019 (COVID-19)	Ongoing global pandemic—to be determined	[11,12,13]

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
