# Peer review of "Enteric Viral Co-Infections: Pathogenesis and Perspective"

_viruses, 2020, doi:10.3390/v12080904_

Round 1

Reviewer 1 Report

Virus-virus coinfection causing AGE is common in clinical patients.

Page 1

The title for table 1: “Summary of the major enteric virus pathogens causing AGE”  

It is not correct. There are viruses that do not cause AGE. Poliovirus is one of viruses via oral-fecal infection and cause poliomyelitis but not gastroenteritis.  Suggesting that to delete “causing AGE”. The size of a viral genome could be also included in table 1, keep it together with the virus genome.

In table 1, Picornaviridae: PV, EV, and CV are old classification. Now it should be Picornaviridae,  Enterovirus genus, enterovirus species A and B (EV and CV); species C (Poliovirus); species D: EV-D68.

Page 4 line 3:  Piconaviridae:  Piconaviruses poliovirus(PV), enteroviruses (EV), and coxsakieviruses(CV). This sentence is wrong. According to new ICTV 2019 version.

Enterovirus, one of genus within Picornaviridae is divided into 15 species from enterovirus species A to L and rhinovirus A-C. Polioviruses belong to species C.

Page 4 line 10: the abbreviation of OPV: what’s a stand for? Oral poliovirus vaccine or Sabin’s vaccine? Please insert it into the text .

Page 4 line 15: Other human non-polio enterovirus including EV-A71, EV-D68, CV-A6 and CVA16 causing hand, foot, mouth diseases.   Wrong sentences.  EV-D68 doesn’t cause HFMD. It cause severe respiratory infection, like rhinovirus infection. “Non-polio enterovirus” or NPEV is wrong words, due to human rhinoviruses are also classified into the enterovirus genus.  

Abbreviation Too many abbreviations in the article, some of them are confusing, for each virus were named differently throughout whole article:

For example: rotavirus: RV, HRV, mRV;  Norovirus: NoV, HuNoV, mNV, Astrovirus: AstV, HAstV, muAstV. Norovirus: presenting NoV for human norovirus but using NV for mouse norovirus. They are not consistency and quite confusing. Please simplify the abbreviation by using only one for one virus: such as rotavirus: RV, and human RV or mouse RV, because the HRV could be called as human rhinovirus that was very commonly used in clinical diagnosis. 

Page 4: Coronaviridae:  Coronavirus was one of virus could cause infection via oral-fecal infection and respiratory tract infection since 20 years ago. Thus, coronavirus, covid19 could be included in the table 1.

Not only Sars-CoV-2 virus but also other coronavirus was reported previously. Covid19: 13 million cases and 570000 death s, please update this information and indicate the time point for the data collection.

Page 7, line 6: “and will continue to be an important consideration as vaccine are developed for other enteric viruses”. Enterovirus (EV) genus contains at least 150 genotypes for enterovirus and 150 for rhinovirus. To develop vaccine for EV will be very difficulty and vaccinate 150 times in one person even more difficulty.

Page 9. Fig. 2

A detail figure legend are needed, including DCV, FHV, and PEDV RECV:  what’s stand for. There is a chimpanzee in the Fig.2 but in text 4.1 Non-human primates:

It is talking about rhesus enteric Calicivirus and simian rotavirus. Thus, it will be better having rhesus monkey in the Fig.2.

Page 11.

In the conclusions, authors should present a clear information about what virus –what virus-coinfection could increase the severity of acute gastroenteritis in patients, or reduce, or no affect on the severity of AGE and duration of symptoms.  Mouse AstV can black mouse NoV infection in vitro model. How about in human enteric viruses infections in patients?  This information is very important for reader to understand the virus-virus-coinfection.

Authors should carefully check the article to make reader easily understand. A clear short conclusion should cover important information about this field.

to those words using full name.

Norovirus: HNoV, MNV

Coronavirus

NPEV Non-polio enterovirus, this sentence was used previously before rhinovirus was classified to enterovirus family, now it is not correctly to be here, due to rhinovirus A, B, C do not cause diarrhea rather than cause respiratory infection and asthma at the age under 3 years old.

 AGE acute gastroenteritis

Abbreviation is too many

Author Response

We are thankful to Reviewer 1 for their helpful critiques. We have addressed the reviewers’ concerns as described below.

  1. The title for table 1: “Summary of the major enteric virus pathogens causing AGE”  

It is not correct. There are viruses that do not cause AGE. Poliovirus is one of viruses via oral-fecal infection and cause poliomyelitis but not gastroenteritis.  Suggesting that to delete “causing AGE”. The size of a viral genome could be also included in table 1, keep it together with the virus genome.

Response: We thank the reviewer for bringing this to our attention and apologize for this oversight. We have corrected the title for Table 1 as well as included the genome sizes for each virus family.

  1. In table 1, Picornaviridae: PV, EV, and CV are old classification. Now it should be Picornaviridae,  Enterovirus genus, enterovirus species A and B (EV and CV); species C (Poliovirus); species D: EV-D68.

Response: We have now updated Table 1 and the relevant text in the review to reflect this correction. We removed EV-D68 since we do not discuss it further and clarified these to be “Major Agents” to indicate our listing is not fully comprehensive.

  1. Page 4 line 3:  Piconaviridae:  Piconaviruses poliovirus(PV), enteroviruses (EV), and coxsakieviruses(CV). This sentence is wrong. According to new ICTV 2019 version.

Enterovirus, one of genus within Picornaviridae is divided into 15 species from enterovirus species A to L and rhinovirus A-C. Polioviruses belong to species C.

Response: We thank the reviewer for identifying this error; we have adjusted the text on Page 4 line 129-130 for the viruses belonging to Picornaviridae family nomenclature as per the new ICTV version.

  1. Page 4 line 10: the abbreviation of OPV: what’s a stand for? Oral poliovirus vaccine or Sabin’s vaccine? Please insert it into the text.

Response: We have updated the text on Page 4 line 137 to clarify OPV as oral polio vaccine.

  1. Page 4 line 15: Other human non-polio enterovirus including EV-A71, EV-D68, CV-A6 and CVA16 causing hand, foot, mouth diseases.   Wrong sentences.  EV-D68 doesn’t cause HFMD. It cause severe respiratory infection, like rhinovirus infection. “Non-polio enterovirus” or NPEV is wrong words, due to human rhinoviruses are also classified into the enterovirus genus.  

Response: We thank the reviewer for bringing this to our attention. We have deleted EV-D68 from the list of enteroviruses causing HFMD on Page 4 line 142. We also removed the term NPEV throughout our manuscript as per the reviewer’s suggestion.

  1. Abbreviation Too many abbreviations in the article, some of them are confusing, for each virus were named differently throughout whole article:

For example: rotavirus: RV, HRV, mRV;  Norovirus: NoV, HuNoV, mNV, Astrovirus: AstV, HAstV, muAstV. Norovirus: presenting NoV for human norovirus but using NV for mouse norovirus. They are not consistency and quite confusing. Please simplify the abbreviation by using only one for one virus: such as rotavirus: RV, and human RV or mouse RV, because the HRV could be called as human rhinovirus that was very commonly used in clinical diagnosis. 

Response: We apologize for the inconsistency in abbreviations. We have now modified the abbreviations throughout the manuscript and in the figures as per the reviewer’s suggestion. For example, we now exclusively use the abbreviation AstV, and specify human or murine as needed.

  1. Page 4: Coronaviridae:  Coronavirus was one of virus could cause infection via oral-fecal infection and respiratory tract infection since 20 years ago. Thus, coronavirus, covid19 could be included in the table 1. Not only Sars-CoV-2 virus but also other coronavirus was reported previously. Covid19: 13 million cases and 570000 death s, please update this information and indicate the time point for the data collection.

Response: We thank the reviewer for this critical comment. We have added the Coronaviridae family of viruses, including SARS and SARS-CoV-2, into Table 1. We have also updated the most recent counts of COVID-19 cases and deaths at the time of submission and added in a timepoint for this data (Page 5, line 169-170).

  1. Page 7, line 6: “and will continue to be an important consideration as vaccine are developed for other enteric viruses”. Enterovirus (EV) genus contains at least 150 genotypes for enterovirus and 150 for rhinovirus. To develop vaccine for EV will be very difficulty and vaccinate 150 times in one person even more difficulty.

Response: We agree with the reviewer that some enteric virus families are highly diverse and vaccine development would be quite challenging. We have clarified that we were referring specifically to some enteric viruses, such as Norovirus, for which vaccine development is potentially practical and is already ongoing, by adding in the phrase “such as NoV” to our statement (Page 7, line 292).

  1. Page 9. Fig. 2

A detail figure legend are needed, including DCV, FHV, and PEDV RECV:  what’s stand for. There is a chimpanzee in the Fig.2 but in text 4.1 Non-human primates:

It is talking about rhesus enteric Calicivirus and simian rotavirus. Thus, it will be better having rhesus monkey in the Fig.2.

Response: We are grateful to the reviewer for bringing this to our attention and have included the full names of the viruses as well as replaced the chimpanzee with a rhesus macaque in Fig. 2.

  1. Page 11. In the conclusions, authors should present a clear information about what virus –what virus-coinfection could increase the severity of acute gastroenteritis in patients, or reduce, or no affect on the severity of AGE and duration of symptoms.  Mouse AstV can black mouse NoV infection in vitro model. How about in human enteric viruses infections in patients?  This information is very important for reader to understand the virus-virus-coinfection.

Response: We agree with the reviewer that this is a key area of interest, and have addressed this topic as best we are currently able to by discussing current data: “Some reports indicate that RV co-infections, at least in healthy children, are not associated with greater symptom severity than single RV infection [22, 24], while another reported that co-infections with NoV may lead to more severe AGE with enhanced comorbidities such as vomiting and fever than NoV alone [94]” on Page 6 line 224-227.

However, most of the reports for human coinfections have been epidemiological surveys without providing substantial insights into the underlying outcomes of coinfection versus infection with a single pathogen. Similarly, there is a significant gap in the data currently available regarding coinfections in animal models. Our manuscript has attempted to highlight that substantially greater work is needed to address this important point (for example, “Additional study of co-infections in both immunocompromised and elderly cohorts may help to provide insights into mechanisms related to immune regulation of enteric viruses” on Page 6 line 245-247), and we have added the following statements, “Further clinical studies exploring the effects of co-infection on symptom severity are needed to better clarify the relative contribution of co-infection to disease outcomes.” and “There is a great remaining need for careful exploration of the effects of co-infection on symptom severity and duration, as well as development of protective immunity, in clinical cohorts.” to the 2.1 section and the conclusion section respectively to further emphasize this point.

  1. Authors should carefully check the article to make reader easily understand. A clear short conclusion should cover important information about this field.

Response: As detailed throughout this manuscript, the virus-virus co-infection field is still in its infancy and much work needs to be done to elucidate a more complete story of enteric viral co-infections. Our conclusion now covers the main aspects discussed in the review, and importantly, it highlights the current state of the field featuring the main gaps in knowledge and potential study models to address these scientific questions.

  1. to those words using full name. Norovirus: HNoV, MNV; Coronavirus;” and “Abbreviation is too many”

Response: Related to reviewer comment #6, we have now modified the abbreviations throughout the manuscript to keep it uniform and to attempt to make it easier to read and understand without excessive abbreviations.

  1. NPEV Non-polio enterovirus, this sentence was used previously before rhinovirus was classified to enterovirus family, now it is not correctly to be here, due to rhinovirus A, B, C do not cause diarrhea rather than cause respiratory infection and asthma at the age under 3 years old.

Response: We have changed this sentence, and removed reference to NPEV throughout our manuscript.

Reviewer 2 Report

The Makimaa manuscript deals with enteric co-infections, a field that is little investigated and therefore highly interesting, especially at the time of the SARS-CoV-2 pandemic. The topic is well covered, but there are some unclear points that should be rewritten, avoiding to treat some topics as a simple list of viral infection rates in different countries (i.e. pg. 4-5). In addition, a hint on enteric co-infections in the elderly would be welcome.

Author Response

Response: We thank the reviewer for this positive assessment of our work. In this review, we strived to provide the most up-to-date information in the field of enteric viral co-infections, including a general sense of the prevalence of enteric virus coinfections identified in different countries across diverse groups. We have added in some additional statements to help summarize our points, including “Further clinical studies exploring the effects of co-infection on symptom severity are needed to better clarify the relative contribution of co-infection to disease outcomes.” and “There is a great remaining need for careful exploration of the effects of co-infection on symptom severity and duration, as well as development of protective immunity, in clinical cohorts.” to the 2.1 section and the conclusion section as mentioned above.

Enteric viral co-infection rates in elderly cohorts have unfortunately been extremely understudied, and we were unable to find any studies specifically reporting the prevalence of co-infections in the elderly. We have briefly discussed enteric viral co-infections in the elderly under section 2.1, on Page 6 line 234-235, and added additional emphasis as to the importance of additional studies specifically examining immunocompromised and elderly populations (Page 6 line 245-247).

Round 2

Reviewer 1 Report

According to my requests, the author made improvements one by one to the revised version of the manuscript.